

# QMetrology from QCosmology:
# Study with entangled two qubit open quantum system
# in De Sitter space

**Sayantan Choudhury**[1,2,3⋆], **Satyaki Chowdhury**[2,3],
**Nitin Gupta**[4] **and Abinash Swain**[2,3]

**1** Quantum Gravity and Unified Theory and Theoretical Cosmology Group, Max Planck
Institute for Gravitational Physics (Albert Einstein Institute),
Am Mühlenberg 1, 14476 Potsdam-Golm,Germany.
**2** National Institute of Science Education and Research,
Jatni, Bhubaneswar, Odisha - 752050, India.
**3** Homi Bhabha National Institute, Training School Complex,
Anushakti Nagar, Mumbai - 400085, India.
**4** Department of Physical Sciences, Indian Institute of Science Education & Research Mohali,
Manauli PO 140306 Punjab, India.

⋆ sayantan.choudhury@niser.ac.in

## Abstract

In this paper, our prime objective is to apply the techniques of *parameter estimation theory* and the concept of *Quantum Metrology* in the form of *Fisher Information* to investigate the role of certain physical quantities in the open quantum dynamics of a two entangled qubit system under the Markovian approximation. There exist various physical parameters which characterize such system, but can not be treated as any quantum mechanical observable. It becomes imperative to do a detailed parameter estimation analysis to determine the physically consistent parameter space of such quantities. We apply both Classical Fisher Information (CFI) and Quantum Fisher Information (QFI) to correctly estimate these parameters, which play significant role to describe the out-of-equilibrium and the long range quantum entanglement phenomena of open quantum system. *Quantum Metrology*, compared to *classical parameter estimation theory*, plays a two-fold superior role, improving the precision and accuracy of parameter estimation. Additionally, in this paper we present a new avenue in terms of *Quantum Metrology*, which beats the classical parameter estimation. We also present an interesting result of *revival of out-of-equilibrium feature at the late time scales, arising due to the long range quantum entanglement at early time scale and provide a physical interpretation for the same in terms of Bell's Inequality Violation in early time scale giving rise to non-locality.*

## 1 Introduction

A quantum system in reality is never considered to be a closed system, it always interacts with the environment no matter how weakly. Understanding the dynamics of a quantum system with the effect of an environment has attracted much attention recently [1–3]. A well studied example is that of the entangled dynamics of two qubits in open quantum system (OQS), described by the weak interaction with a massless probe scalar field, playing the role of thermal bath or environment [4–7, 10, 11]. The time evolutionary picture of such an OQS is described by the adiabatic interactions between the system under consideration (which is in our context the two qubit entangled system) and its thermal environment and is non unitary. The non-unitary time evolution of such a system is appearing as an outcome of quantum dissipative effects which can be explicitly obtained by solving the effective master equation, also known as the *Gorini-Kossakowski-Sudarshan-Lindblad (GSKL) master equation*, expressed in terms of the reduced density matrix obtained by tracing out the unwanted bath degrees of freedom from the total system [1]. The non-unitarity in the evolution process is mainly controlled by the *Lindbladian operator* which is primarily responsible for introducing quantum mechanical dissipation into the system due to the interaction with the environment.

The environmental interaction is the main culprit in this discussion which spoils the unitary time evolution of the physical system of interest under consideration. Therefore it becomes essential to develop methods for accurately estimating the parameters of the theory which directly controls the influence that the environment has on the physical system under consideration. In the present context of discussion it is often called the *coupling parameter*. In the study of any physical system to model the out-of-equilibrium [5, 12–14] scenario, it becomes crucial to have an estimation of the time at which the out-of-equilibrium feature starts expressing in the system and the time at which the system finally equilibrates with the environment. This evolutionary time scale of the physical system under the influence of the thermal bath provides an approximate estimation of the strength of the coupling between the

---

[1] Technically the partial trace operation in the present context is identified as the path integration operation over the bath degrees of freedom

physical system and the surroundings. Study of open quantum systems has been done both in flat and curved spacetime [6–8,15,16] in many different contexts. Various models have been constructed to realise an open quantum system and study its various properties. Specifically, models consisting of one or two qubits with scalar fields acting as the thermal bath has been extensively studied [6–8,15,16]. Studies involving two entangled qubits in different classical gravitational background uses the Resonant Casimir Polder Interaction (RCPI) arising from the vaccum fluctuations of quantum fields, between the two entangled qubits. It gave a new way of extracting information about spacetime curvature from casimir physics [8,9].

The theory of *Quantum Information Processing* (QIP) or *Parameter Estimation Theory* (PET) plays a pivotal role in the context of *quantum information and computation*. The study of estimating parameters forms the subject matter of the science of estimation theory. Since the advent of quantum physics in early 1900s it is imperative that all careful and precision measurement experiments be necessarily of a quantum nature. This comes under the purview of *Quantum Metrology*, which is the study of performing high-precision measurements and estimations with the promise of delivering techniques which outperform their classical counterparts. Statistical errors form a part and parcel of any measurement process. Protocols based on the ideas of *Quantum Metrology* can help one achieve precision levels which surpass their classical counterparts and significantly reduce statistical errors. *Quantum Metrology* is able to saturate the *Heisenberg Limit* [17,21] which specifies how precision of a measurement scales with variation of energy. For example, in case of interferometers, using classical measurement protocols one is limited to the shot-noise limit of $(\Delta \xi)^2 \geq 1/N$ where $\xi$ is some physical parameter to be estimated and $N$ is the number of photons. *Quantum Metrology* techniques allow one to reach the *Heisenberg Limit* of $(\Delta \xi)^2 \geq 1/N$. Additionally, by it's very construction, *Quantum Metrology* involves estimation of physical parameters which have classical counterparts as well as estimation of those which are purely quantum mechanical in origin. Hence, *Quantum Metrology* not only refines *precision measurement* but it also provides fresh avenues of probes which are otherwise inaccessible. The technique of fisher information has previously been used in various different models for studying the dynamics of systems in curved spacetime [18–20]

The precision of parameter estimation is measured by *Fisher information* [22]. In the field of parameter estimation, the prime focus is to give an estimate of the values of the unknown physical parameters labelling a quantum mechanical model and to enhance the precision of resolution. *Quantum Fisher Information* (QFI), considered as another version of *Skew Information* [23] which is considered as one of the most important measures in the context of PET. It measures the sensitivity or the response of a system with respect to changes in the parameters that governs the information regarding the physical system under consideration. Recent studies of QFI has shown its enormous applicability in other fields apart from PET [24–41]. It also acts as a resource to detect the quantum entanglement and its long range effect among qubits [42–44]. In ref. [45], the authors recently proposed an experimental scheme to quantify the lower bounds of *Fisher Information*.

In this paper, we use *Fisher Information* to investigate the minimal evolutionary time scale between the two distinguishable quantum states of the entangled two qubit system, which basically represents the time scale at which out-of-equilibrium phenomenon starts appearing in the system due to its interaction with the thermal bath and the time scale when the system finally reaches the thermal equilibrium state of the bath. Apart from this, *Quantum Fisher Information* (QFI) can also be used to determine the interaction or coupling strength with which the bath influences the system and to provide a physical justification for considering *Markovian approximation* of the environment during our analysis.

The plan of the paper is as follows:- In section : The Two Qubit Open Quantum System, we describe our model of two entangled qubits in interaction with the thermal bath and the

characteristics of such a model resembling an open quantum system. In section :Quantum Fisher Information, we discuss the basics and the derivation of the expression of the fisher information. In Bloch Vector representation of Fisher Information, we provide the general Bloch vector representation of the eigenvalues and eigenvectors of the density matrix that characterizes our two qubit reduced subsystem. Finally, in section Estimation of Parameters : Estimation of TimeScale, Estimation of Euclidean Distance, Estimation of Coupling Strength we apply the techniques of *Fisher Information*, both Classical and Quantum to estimate some of the essential physical parameters that plays a pivotal role in determining the time evolutionary dynamics of the system under consideration. We end with some essential conclusions obtained from this analysis and provide some of the future prospects where Quantum Information science can be used as an essential probe to describe some physical phenomena in the context of Cosmology described by an Open Quantum System.

## 2   The Two Qubit Open Quantum System

For our work, we review a model of two identical entangled qubits as described with much clarity in [5]. Each of the qubits have two internal energy levels. The considered system is conformally coupled to a massless scalar field in the static De-Sitter space-time in $3 + 1$ dimensions. The interaction between the two identical qubit system and the bath is assumed to be weak and perfectly consistent with the underlying requirement of perturbation theory. The system of two entangled OQS is represented by the following Hamiltonian:

$$H_T = H_S \otimes I_B + I_S \otimes H_B + H_I, \tag{1}$$

where $H_T$ is the total Hamiltonian of the entire configuration of system and bath. $H_S, H_B, H_I$ represent the system, bath and interaction Hamiltonian respectively. Also, $I_S$ and $I_B$ represent identity operators of the system and bath respectively and it is used to describe the absence of system and bath during the quantification of the Hamiltonian of the bath and system solely generated from the self interactions. The parameter $t$ appearing in this context is the conformal time and is given by $t = \left( \sqrt{1 - r^2/\zeta^2} \right) t'$, where $\zeta = \sqrt{3/\Lambda}$, $\Lambda > 0$ for $3 + 1$ dimensional static de Sitter space and $t'$ is the physical time[2].

The system of two entangled qubits is described by the linear combinations of the contributions coming from the individual qubit and is described by the following Hamiltonian:

$$H_S = \frac{\omega}{2} \sum_{\alpha=1}^{2} \hat{n}^\alpha \cdot \vec{\sigma}^\alpha, \tag{2}$$

where $\omega$ represents the renormalized energy level for two atoms, given by:

$$\omega = \begin{cases} \omega_0 + i[\mathcal{K}^{(11)}(-\omega_0) - \mathcal{K}^{(11)}(\omega_0)] & \textbf{Atom 1} \\ \omega_0 + i[\mathcal{K}^{(22)}(-\omega_0) - \mathcal{K}^{(22)}(\omega_0)] & \textbf{Atom 2} \end{cases} . \tag{3}$$

In this construction, $\omega$, $\omega_0$ and the factor $k$ which is appearing in the Fourier transform of the Wightman functions (see appendix) all are taken real to perform the Fisher Information analysis in this paper and this is necessarily required to suffice the present purpose. In this connection additionally it is important to note that, we have used $k\omega \gg 1$ in the Fourier transform of the Wightman functions which we have used during our analysis. Also, $\mathcal{K}^{\alpha\alpha}(\pm\omega_0)$ for $\alpha \in \{1, 2\}$ are Hilbert transformations of two-point Wightmann functions which after the detailed computations will turn out to be imaginary and the whole combination stated in

---

[2]This is atypical because in literature use $\tau$ as the conformal time and $t$ as the physical time.

equation 3 ultimately makes the renormalized frequency $\omega$ real as we have previously stated $\omega_0$ is also real strictly [3]. $\hat{n}^{\alpha}$ represents the arbitrary orientation of the individual qubit and $\vec{\sigma}$ is represented by the three basis vectors $\vec{\sigma} := (\sigma_+, \sigma_-, \sigma_3)$ where $\sigma_{\pm}$ is defined as:

$$\sigma_{\pm} := \frac{1}{\sqrt{2}}(\sigma_1 \pm \sigma_2), \tag{4}$$

where $(\sigma_1, \sigma_2, \sigma_3)$ are the Pauli matrices. Since we are considering a new transformed basis to represent the Pauli matrix vector, we will carry forward this convention for the rest of the computation of this paper. [4].

The massless free rescaled field $\Phi$ acting as the thermal bath is described by the following Hamiltonian:

$$H_B = \int_0^{\infty} dr \int_0^{\pi} d\theta \int_0^{2\pi} d\phi \left[ \frac{\Pi_{\Phi}^2}{2} + \chi(r, \theta, \phi) \right], \tag{5}$$

where we define the function, $\chi(r, \theta, \phi)$, as:

$$\chi(r, \theta, \phi) = \frac{r^2 \sin^2 \theta}{2} \left\{ r^2 (\partial_r \Phi)^2 + \frac{(\partial_\theta \Phi)^2 + \frac{(\partial_\phi \Phi)^2}{\sin^2 \theta}}{(1 - \frac{r^2}{\alpha^2})} \right\}. \tag{6}$$

Here $\Pi_{\Phi}$ is the canonically conjugate momenta of the field $\Phi$.

The Hamiltonian of the interaction part between the entangled two qubit system and the massless scalar field $\Phi$ placed at the thermal environment is characterized by:

$$H_I(t) = \mu \sum_{\alpha=1}^{2} \underbrace{\underbrace{\frac{\omega}{2}(\hat{n}^{\alpha} \cdot \vec{\sigma}^{\alpha})}_{\text{\textcolor{red}{Individual Qubit System}}} \underbrace{\Phi(x^{\alpha})}_{\text{\textcolor{red}{Bath}}}}_{\text{\textcolor{blue}{System−Bath interaction via qubit index } \alpha}}. \tag{7}$$

Here we assume that the interaction strength $\mu$ to be very small so that the perturbation techniques can be used. Also it is important to mention that, since we are considering identical qubits for this analysis, it is expected to have same coupling strengths for each of them with the massless probe scalar field $\Phi$. For the more complicated situation one may consider, different coupling strengths, however in this work we have not considered such possibilities for the sake of simplicity.

The non-unitary time evolution of such an OQS is governed by the folowing GSKL master equation:

$$\frac{d}{dt}\rho_S(t) = -i[H_{eff}(t), \rho_S(t)] + \mathcal{L}[\rho_S(t)], \tag{8}$$

where $\rho_S(t) = \text{Tr}_B \rho_T(t)$ is the reduced density matrix of the system with $\rho_T$ being the total density matrix of the entire configuration. Here $\text{Tr}_B$ is the partial trace operation over the bath

---

[3]It is important to note that, in our previous work [4], we have used an additional condition $\coth(\pi k \omega_0) = 0$ to simplify the expressions for the derived expressions for the spectral shifts. This condition we have not strictly used in this paper to perform the Fisher Information analysis using the density matrix, as the present analysis can only be performed in presence of all real parameters.

[4]The new basis of the Pauli matrix vector resembles the light cone gauge which is commonly used in the context of gauge theories and is used to remove the ambiguities appearing from the gauge symmetries. The only difference is, in this kind of gauge choice, either the + or the − component is fixed to be zero and treated as the gauge condition. But in the present context we are not using the basis transformation like the light cone gauge. So the basis transformation in the present context can be treated as the extension of the light cone (or null) coordinate transformation using which one can transform the Pauli matrix vector in a new redefined basis, which simplifies further computations. Additionally, it is important to point here that, after introducing this new basis all the physical observables computed from the present open quantum system set up will remain unchanged, only it will help us to perform the computations in a simpler way.

degrees of freedom. This is nothing but applying the path integral operation over the massless bath field $\Phi$ when we represent everything in the language of constructing an effective action for the two qubit system.

$H_{eff}$ is the effective Hamiltonian of the two atomic system, which incorporates the effect of inter atomic interaction aka Resonant Casimir Polder Interaction(RCPI). Also, the last term in the above mentioned evolution equation is known as the Lindbladian, which describes the dissipative contribution due to the influence of the thermal bath on the two entangled atomic system. In the appendix, we discuss about the effective Hamiltonian and the Lindbadian with greater detail.

To have a better understanding of the system and to estimate the parameters, we must solve the GSKL Master Equation. For this purpose, we parametrize our arbitrary two qubit subsystem density matrix in terms of Pauli matrices by the following expression:

$$\rho_S(t) = \frac{1}{4} \sum_{p,q=0,+,-,3} a_{pq}(t)\, \sigma_p \otimes \sigma_q, \tag{9}$$

where the time dependent expansion coefficients are fixed from the solution of the GSKL master equation subject to the boundary condition applicable at the large time limiting situation which corresponds to the thermal equilibrium.

For the sake of convenience, we have used $\sigma_+, \sigma_-$ and $\sigma_3$ along with $\sigma_0$ (identity) to express the density matrix in terms of Bloch vector components. In terms of the Bloch vector representation, density matrices of two qubits are represented by:

**Qubit 1** :

$$\rho_1(t) \quad := \quad \frac{1}{2}(1 + \mathcal{A}(t).\boldsymbol{\sigma}) = \frac{1}{2}\left(1 + \sum_{i=+,-,3} \mathcal{A}_i(t)\sigma_i\right), \tag{10}$$

**Qubit 2** :

$$\rho_2(t) \quad := \quad \frac{1}{2}(1 + \mathcal{B}(t).\boldsymbol{\sigma}) = \frac{1}{2}\left(1 + \sum_{j=+,-,3} \mathcal{B}_j(t)\sigma_j\right). \tag{11}$$

Consequently, for the combined two qubit system the density matrix is represented by the following expression:

$$\rho_S(t) = \rho_1(t_1) \otimes \rho_2(t_2) = \frac{1}{4} \sum_{i,j=0,+,-,3} a_{ij}(t)\, \sigma_i \otimes \sigma_j, \tag{12}$$

where we define:

$$a_{00}(t): \quad = \quad 1, \tag{13}$$
$$a_{i0}(t): \quad = \quad \mathcal{A}_i(t), \tag{14}$$
$$a_{0i}(t): \quad = \quad \mathcal{B}_i(t), \tag{15}$$
$$a_{ij}(t): \quad = \quad \mathcal{A}_i(t)\mathcal{B}_j(t). \tag{16}$$

In ref. [4], the authors have also considered a similar two qubit system which is interacting with the free massless scalar field acting as the thermal bath. To describe the system, the authors have used a similar Bloch vector representation as given in equation 9. Substituting equation 9 in the *GSKL master equation*, the time dependence of the *Bloch vectors* and hence, the time dependence of the sub system density matrix can be calculated. The soultions to Master equation are provided in [4]. In this paper, we use these solutions to estimate some of the essential physical parameters that plays an essential role in studying the out-of-equilibrium as well as the equilibrium properties of such an entangled sub system in OQS.

# 3  Quantum Fisher Information (QFI)

One can argue that precision measurement forms a roadmap to better technologies and new physical phenomena. It is possible to model a physical system in terms of parameters that can be estimated to extract information about the relevant physical system. *Fisher Information*(FI) forms the *crux of Metrology*, be it classical or quantum mechanical in nature. In this section we succinctly give a brief review of Quantum Fisher Information(QFI).

Fisher Information measures the changes in states of a physical system with respect to *a parameter* or *a family of parameters* i.e. *a parameter vector*. For classical statistical systems, the states are represented as probability distributions whereas in quantum mechanical systems the states are characterized as density matrices. The connection between FI modelled through a parameter (or estimator) and the variance of that estimator is established through the *Cramér-Rao Bound* as an inequality which limits the precision of measurement, thereby making FI a cornerstone in the study of *Quantum Metrology*.

In the following section, we provide a brief review on *Quantum Fisher Information* (QFI) and defer the interested readers to more careful and extensive studies. *Quantum Fisher Information Matrix* (QFIM) plays a pivotal role in *Quantum Information Theory* and *Quantum Metrology* by improving the accuracy of parameter estimation especially of those parameters which are difficult to measure in principle or lie beyond experimental capability. The precision of determining a parameter is given by QFI; larger the QFI, higher the precision. We will use QFI as an estimator of parameters involved in the dynamics of the evolution of the system. Below we arrive at a formula for QFI(M) that we are going to use in estimating the parameters in our theory.

Let $\vec{\xi}$ be the parameter encoded in a quantum state, in other words our density matrix is a function of $\vec{\xi}$ i.e $\rho = \rho(\vec{\xi})$. Then QFIM is defined as:

$$\mathcal{F}_{ab} = \frac{1}{2} Tr(\rho\{L_a, L_b\}), \tag{17}$$

where $L_a$ denotes the symmetric logarithmic derivative (SLD) of the parameter $\xi^a$ as follows:

$$\partial_a \rho = \frac{1}{2}(\rho L_a + L_a \rho), \tag{18}$$

with $\partial_a \equiv \frac{\partial}{\partial \xi^a}$ denoting the partial differentiation with respect to the desired parameter (here, $\xi^a$). In this context, $\mathcal{F}_{ab}$ forms a matrix called Quantum Fisher Information Matrix (QFIM) and the diagonal elements of this matrix are our so called, *Quantum Fisher Information* (QFI).

The diagonal elements for the matrix 17 are given as:

$$\mathcal{F}_{aa} = Tr(\rho L_a^2). \tag{19}$$

The parameters can be encoded to a quantum state mainly through the dynamics. Sometimes the parameter is encoded in the Hamiltonian of the system itself and sometimes it may be encoded through the interaction with the surrounding and sometimes through both. In this paper, we only consider the latter where the parameter arises because of the interaction of system with the bath.

Typical derivations of QFIM assume a full-ranked density matrix *i.e.* all the eigenvalues of the density matrix are positive. This special type of density matrix can be written as:

$$\rho = \sum_{i=0}^{D-1} \lambda_i |\lambda_i\rangle \langle \lambda_i|, \tag{20}$$

where the eigenvalues $\lambda_i > 0$ and $|\lambda_i\rangle$ are the corresponding eigenvectors. *D* is the dimension of $\rho$.

After substituting the spectral decomposition of the density matrix into equation 17 and 18 and using the completeness property which is given by:

$$\mathbb{1} = \sum_{i=0}^{D-1} |\lambda_i\rangle \langle\lambda_i|, \tag{21}$$

the QFIM for such a full rank density matrix can be obtained as follows:

$$\mathcal{F}_{ab} = \sum_{i,j=0}^{D-1} \frac{2\Re(\langle\lambda_i| \partial_a\rho |\lambda_j\rangle\langle\lambda_j| \partial_b\rho |\lambda_i\rangle)}{\lambda_i + \lambda_j}, \tag{22}$$

where $\Re$ denotes the *Real* part of a complex number. The *Support* of a finite dimensional density matrix can be given as:

$$S = \{\lambda_i \in \{\lambda_i\} | \lambda_i \neq 0\}.$$

In this case, the spectral decomposition of density matrix is given as

$$\rho = \sum_{\lambda_i \in S} \lambda_i |\lambda_i\rangle \langle\lambda_i|.$$

For this case, the QFIM is given by the following expression:

$$\mathcal{F}_{ab} = \sum_{\lambda_i \in S} \frac{(\partial_a\lambda_i)(\partial_b\lambda_i)}{\lambda_i} + \sum_{\lambda_i \in S} 4\lambda_i \Re(\langle\partial_a\lambda_i|\partial_b\lambda_i\rangle) - \sum_{\lambda_i,\lambda_j \in S} \frac{8\lambda_i\lambda_j}{\lambda_i + \lambda_j} \Re(\langle\partial_a\lambda_i|\lambda_j\rangle\langle\lambda_j|\partial_b\lambda_i\rangle), \tag{23}$$

and hence the QFI can be expressed as:

$$\mathcal{F}_{aa} = \sum_{\lambda_i \in S} \frac{(\partial_a\lambda_i)^2}{\lambda_i} + \sum_{\lambda_i \in S} 4\lambda_i\langle\partial_a\lambda_i|\partial_a\lambda_i\rangle - \sum_{\lambda_i,\lambda_j \in S} \frac{8\lambda_i\lambda_j}{\lambda_i + \lambda_j} (|\langle\partial_a\lambda_i|\lambda_j\rangle|^2). \tag{24}$$

Therefore, the expression of Fisher Information, $\mathcal{F}_I$, can be rewritten with re-identification of terms as

$$\mathcal{F}_I = \mathcal{F}_c + \mathcal{F}_q - \mathcal{F}_m, \tag{25}$$

where $\mathcal{F}_c$ and $\mathcal{F}_q$ respectively represent the classical and quantum part of the Fisher Information of all pure states. The third term $\mathcal{F}_m$ usually arises from the mixture of the first two terms. Explicitly these contributions are written as:

$$\mathcal{F}_c = \sum_{i=1}^{D-1} \frac{1}{\lambda_i} \left(\frac{\partial\lambda_i}{\partial x_a}\right)^2, \tag{26}$$

$$\mathcal{F}_q = 4\sum_{i=1}^{D-1} \lambda_i \left(\left\langle \frac{\partial\lambda_i}{\partial x_a} \Big| \frac{\partial\lambda_i}{\partial x_a} \right\rangle - \left|\left\langle \lambda_i \Big| \frac{\partial\lambda_i}{\partial x_a} \right\rangle\right|^2\right), \tag{27}$$

$$\mathcal{F}_m = 8\sum_{i\neq j}^{D-1} \frac{\lambda_i\lambda_j}{\lambda_i + \lambda_j} \left|\left\langle \lambda_i \Big| \frac{\partial\lambda_i}{\partial x_a} \right\rangle\right|^2. \tag{28}$$

Hence, for a pure state we only have the first two terms of the Fisher Information while for a mixed state the third term needs to be subtracted. From the above argument it is clear that the Fisher Information of a pure state is generally greater than that of a mixed state.

## 3.1 Bloch Vector Representation Of Fisher Information

In this section we provide a general expression for the eigenvalues and eigenvectors of the density matrix for any arbitrary two qubit system expressed in $\sigma_+, \sigma_-, \sigma_3$. The significance of calculating the eigenvalues and eigenvectors lies in parameter estimation using Fisher Information, where the derivatives of the eigenvalues and the eigenvectors are taken with respect to the parameter to be estimated. The eigenvalues of the density matrix in terms of the Bloch vectors can be written as

$$
\begin{aligned}
\lambda_1 &= \frac{1}{4}\left(1 - a_{33}(t) - \mathcal{X}(t)\right), \quad \lambda_3 = \frac{1}{4}\left(1 + a_{33}(t) - \mathcal{Y}(t)\right), \\
\lambda_2 &= \frac{1}{4}\left(1 - a_{33}(t) + \mathcal{X}(t)\right), \quad \lambda_4 = \frac{1}{4}\left(1 + a_{33}(t) + \mathcal{Y}(t)\right).
\end{aligned}
\tag{29}
$$

Similarly, the eigenvectors in terms of the Bloch vector components can be written as

$$
\begin{aligned}
|\lambda_1\rangle &= \left(0, -\frac{\sqrt{a_{+-}(t)}}{\sqrt{a_{-+}(t)}}, 1, 0\right), \quad |\lambda_3\rangle = \left(\frac{2a_{03}(t) - \mathcal{Y}(t)}{a_{--}(t)}, 0, 0, 1\right), \\
|\lambda_2\rangle &= \left(0, \frac{\sqrt{a_{+-}(t)}}{\sqrt{a_{-+}(t)}}, 1, 0\right), \qquad |\lambda_4\rangle = \left(\frac{2a_{03}(t) + \mathcal{Y}(t)}{a_{--}(t)}, 0, 0, 1\right),
\end{aligned}
\tag{30}
$$

where $\mathcal{X}(t)$ and $\mathcal{Y}(t)$ represented by:

$$
\mathcal{X}(t) := \sqrt{a_{-+}(t)a_{+-}(t)},
\tag{31}
$$

$$
\mathcal{Y}(t) := \sqrt{4a_{03}^2(t) + a_{--}(t)a_{++}(t)}.
\tag{32}
$$

# 4 Estimation of Parameters

Estimation of a parameter is associated with it's *(C/Q) Fisher Information* and the quality of estimation is established through the *(C/Q)Cramér-Rao Bound*. To estimate a parameter independently of other parameters we need to obtain the corresponding diagonal element of *Quantum Fisher Information Matrix* (QFIM) which consists of three terms : $\mathcal{F}_c, \mathcal{F}_q, \mathcal{F}_m$ as discussed in the preceding section. This involves taking derivatives of the eigenvalues & eigenvectors of the density matrix with respect to the parameter of our interest.

In this paper we have mainly focused on estimating parameters, which are the most significant ones in characterizing the non equilibrium behavior of the system in the presence of a bath. We also take into account the parameter which determines the degree of entanglement between the two qubits constituting our system. The influence of the bath on the evolution of the system is also a significant consideration in any phenomenological study of an open quantum model. For that purpose we have also taken into account the parameter which determines the magnitude of the influence that the system has on the environment.

## 4.1 Estimation of Timescale

This section primarily focuses on estimating the time scale at which non equilibrium behaviour starts appearing in the system and the time scale at which the system thermally equilibrates with the bath. In any open quantum system model, non equilibrium behaviour of the system is inevitable. So it becomes very crucial to have an estimation of the time at which the system goes out of equilibrium and the time at which it finally equilibrates. This indirect way of estimation can prove to be very useful if one wants to perform an experiment to study the non equilibrium as well as the equilibrium behaviour of such an open quantum system separately.

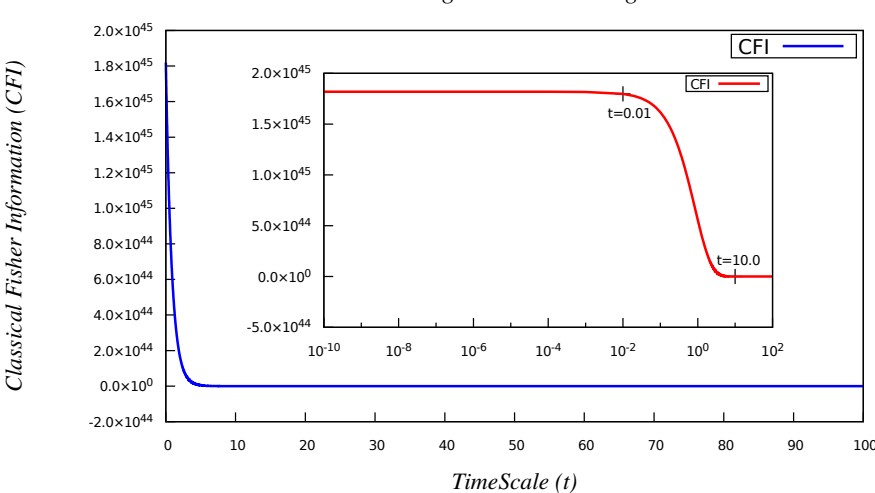

Figure 1: Studying behavior of CFI with changing timescale (t). To study the dependence on timescale independently we have fixed : {$\mu$=0.001, k=0.001, $\omega$=1, $\omega_0 = \frac{i}{0.002}$, L=1, $\tau = 100$}. Relevant points are marked on the plot.

In Fig.1 we have studied the variation of *Classical Fisher Information* (CFI) with varying time scale. According to *Cramér-Rao Bound* the CFI for a parameter is inversely proportional to the variance of that parameter. This allows us to check the maximum value of CFI and select the corresponding time scale as the best estimated time scale by our analysis which turns out to be of the order $10^{-2}$ in our case.

We observe that the CFI remains constant up to $t = 10^{-2}$ and then there is an exponential decrease in it until around $t = 10$ after which is becomes comparatively vanishing. In physical terms this means that,
for our specific model, the two qubits do not start interacting with the background field until about $t = 10^{-2}$ after which they undergo non-equilibrium evolution under interaction with the background field until around $t = 10$ after which the two qubits attain equilibrium with the background field. The dynamics between $t = 10^{-2}$ and $t = 10$ is described according to the principles of open quantum systems applied to a cosmological background in de Sitter Space, i.e., *Open Quantum Cosmology*. This has allowed us to determine the time scales involved in the model and establish the contrast between equilibrium regime and non-equilibrium regime which enables one to do many interesting studies in the non-equilibrium regime. It is to be noted that after $t = 10$ we cannot extract any information out of this system as the CFI becomes comparatively negligible. This posits that to extract useful information out of this system one needs to do all the analysis and experiments in the region $0.01 \leq t \leq 10$ and that this is the region in which information is lost due to interaction with the background field.

In Fig.2 we have studied the variation of Quantum Fisher Information (QFI) with varying time scale. As discussed previously, we look for the value of time scale which yields the largest value of QFI and that time scale is our best estimated time scale using QFI.

It is observed that in the range of time scale : $10^{-10} \leq t \leq 10$, the behaviour of QFI is roughly the same as that of CFI. From $t = 10^{-10}$ to $t = 10^{-2}$ it is almost constant with some fluctuations and after $t = 10^{-2}$ we can observe an exponential decay in the QFI as was observed for CFI. Again, the estimated time scale when the interaction between qubits and environment starts is around $t \sim 10^{-2}$ with non-equilibrium interaction continuing up to around $t = 10$. It is after this region that QFI shows interesting and different behaviour from CFI. Whereas CFI continues to be negligible we get a prominent peak in QFI at $t = 70.11$ which is the unique

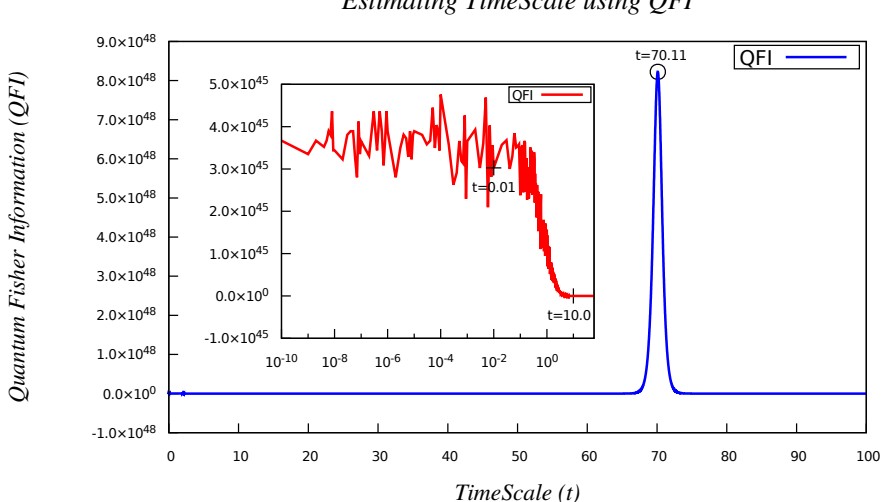

Figure 2: Studying behavior of QFI with varying timescale (t). To study the dependence on timescale independently we have fixed : $\{\mu=0.001, k=0.001, \omega=1, \omega_0 = \frac{i}{0.002}, L=1, \tau = 100\}$. Relevant points are marked on the plot.

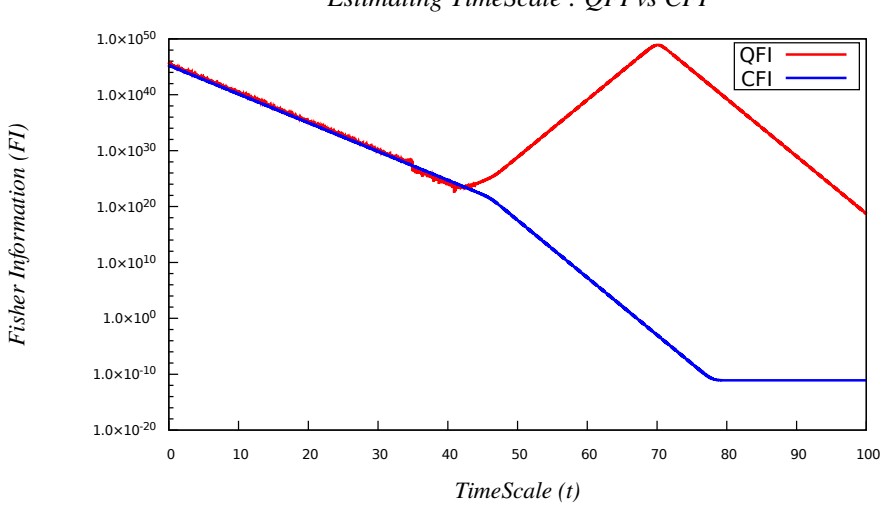

Figure 3: Comparative study of CFI and QFI with changing timescale (t). To study the dependence on timescale independently we have fixed : $\{\mu=0.001, k=0.001, \omega=1, \omega_0 = \frac{i}{0.002}, L=1, \tau = 100\}$

feature present in QFI. This demonstrates that one must use QFI to estimate time scale instead of CFI. Even though their prediction for the non-equilibrium interaction time scales are roughly same, CFI fails to capture the "revival" of information that QFI shows at $t = 70.11$. It is to be noted that the order of magnitude of CFI and QFI is roughly the same up to around $t = 10$ which means that one can use either CFI or QFI for studying time scales in this region but above this region one must rely exclusively on QFI to obtain all the interesting physics.

Finally, to highlight the contrast between CFI and QFI for time scale we have shown a comparative study between them in Fig.3. It is observed that until about $t = 40$, CFI and QFI have the same behavior and then they draw apart. While CFI quickly goes to equilibrium values the QFI rises to give a peak at $t = 70.11$ which indicates that there is a "revival" of the

information profile and that non-equilibrium phenomena can be detected in the approximate range of : $67 \leq t \leq 73$. This provides an interesting and new avenue for the study of non-equilibrium phenomena which is specific to studies which consider QFI.

This motivates one to study for late time non-equilibrium phenomena in entangled systems in cosmological deSitter background where the non-equilibrium phenomena in the late time regime are appearing due to the entanglement generated between system and environment at early times. We provide a detailed physical analysis of this interesting feature as follows.

In Quantum Mechanics, one of the important aspects is to generate long range quantum correlation functions at the late time scale but it is extremely complicated to generate these kinds of correlations in the late time scale regime. The necessary ingredient to have such kind of correlations in Quantum Mechanics is the phenomenon of Quantum Entanglement. But using the usual Quantum Entanglement set up within the framework of Quantum Mechanics it is impossible to generate long range correlations at late time scale.

The only way to achieve the same is to encode non-locality in the correlations in the early time scale, which can be established through Bell's Inequality violation. In our earlier work [46, 47], we have established how one can violate Bell's inequality within the framework of Quantum Mechanics.

So, one can interpret the "revival" in the QFI at late time scale quantum correlations which are appearing as an outcome of non-local initial entanglement in the early time scale. On the other hand, this type of revival also gives information about the out-of-equilibrium phenomena at the late time scale so within our set up the non-locality in the initial correlation is actually connected to the out of equilibrium feature at the late time scale. Additionally, it is important to note that, though it is true that the revival of out-of-equilibrium profile is coming from the time evolution of the initial non-local quantum entanglement we don't exactly know the dynamical equation that is satisfied by the information from early time scale to the late time scale. But the important thing is that these results allow one to confidently state the existence of long range out-of-equilibrium phenomena at the late time scale which trace their origins to the non local Bell's Inequality violation in early time scale i.e. *one can actually establish a connection between non-local quantum entanglement phenomena at the early time scale with the long range out of equilibrium phenomena at late time scale.*

Additionally, we can extend this analysis further in a very interesting direction. Note that when we provide an initial condition at very early time scale the system goes to a out-of-equilibrium phase but to get information about this random phase one needs to explicitly compute the quantum correlation functions, but by itself the computation of these functions are extremely complicated to perform in the quantum regime. In ref. [48], using the principles of Random Matrix Theory the authors have given an explicit computation of these quantum correlation function in the framework of cosmology. Recently in ref. [12], one of the authors tried to give more elegant method of this computation in terms of computing out-of-time-ordered correlation functions within the framework of primordial cosmology. In this context if we wait for a large enough time then the quantum system under consideration equilibriates and we get an estimation of the corresponding equilibrium temperature associated with the thermal bath, which we have actually modelled with massless free scalar field in OQS. So it is expected from our analysis that when the system reaches the thermal equilibrium then from the saturation limiting value of the quantum correlation function one can quantify the upper bound on a measure of quantum randomness, which these days is identified as the *quantum Lyapunov exponent*, $\lambda$, which is the quantum generalization of the *classical Lyapunov exponent* appearing in the classical chaotic dynamical systems. For this reason if we look into the complete picture of the spectrum starting from the initial point when the system is perturbed upto the point when the system gets saturated in terms of quantum correlations, the four point functions, $\langle \Phi(t_1)\Pi_\Phi(t_2)\Phi(t_1)\Pi_\Phi(t_2)\rangle_\beta \sim F(t_1 - t_2)\, e^{\lambda(t_1+t_2)/2}$ growth factor, where $\lambda \leq 2\pi/\beta$

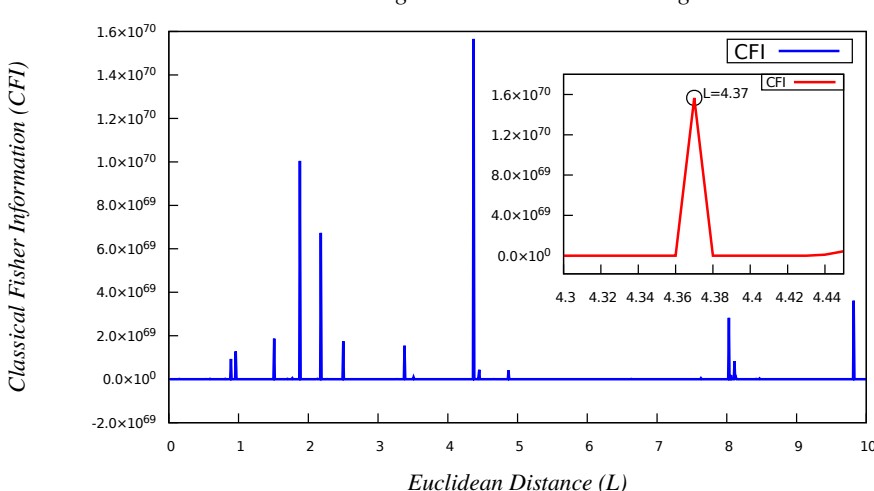

Figure 4: Behavior of CFI with varying Euclidean Distance(L) between the qubits evaluated at : {$\mu$=0.001, k=0.001, $\omega$=1, $\omega_0 = \frac{i}{0.002}$, t=1, $\tau = 100$}. Relevant points are marked in the plot.

is commonly known as *Maldacena-Shenker-Stanford bound* [49], here $\beta = 1/T$ with $T$ being the equilibrium temperature of the thermal bath and $\langle \cdot \rangle_\beta$ represents the thermal expectation value. On the other hand, from the other two quantum correlators, $\langle \Phi(t_1)\Phi(t_2)\Phi(t_1)\Phi(t_2) \rangle_\beta$ and $\langle \Pi_\Phi(t_1)\Pi_\Phi(t_2)\Pi_\Phi(t_1)\Pi_\Phi(t_2) \rangle_\beta$ one can study the random but non-chaotic behaviour of correlation functions.

## 4.2 Estimation of Euclidean Distance

In this section we try to provide an estimation of the distance between two qubits in the static patch of De-sitter space using both classical and quantum Fisher Information. We provide reasons why Quantum Fisher Information is a better measure for estimating any parameter than Classical Fisher Information. This analysis is essential to have a pre-determined idea that an experimentalist, preparing a set up to study entanglement related phenomenon, should have.

In Fig.(4), we have explicitly shown the behavior of Classical Fisher information of our two atomic open quantum system in static patch of de-Sitter space with respect to the euclidean distance between the two qubits. In this analysis we have fixed the value of the other parameters. From the plot it can be seen that the Classical Fisher Information predicts some particular values of the euclidean distance where entanglement between the qubits will be most prominent. This can be seen from the peaks of the plots. However for a particular value of euclidean distance(for these set of parameter values),around 4.37 the maximum of the plot occurs suggesting it to be the most appropriate euclidean distance to study various entanglement process like entropies etc. Thus Classical Fisher Information estimates the euclidean distance between the qubits to be around **4.37** up to certain order of accuracy for these particular choice of other parameter values. In Fig.(5), we have explicitly shown the behaviour of Quantum Fisher information of our two qubit open quantum system in static patch of De-Sitter space with respect to the euclidean distance between the two qubits. The plot shows a peak around 4 which is very close to the estimated value from the Classical Fisher Information. However Quantum Fisher Information shows peaks for larger values of euclidean distance which could not be probed by Classical Fisher Information. The maximum of the plot shows that QFI predicts a

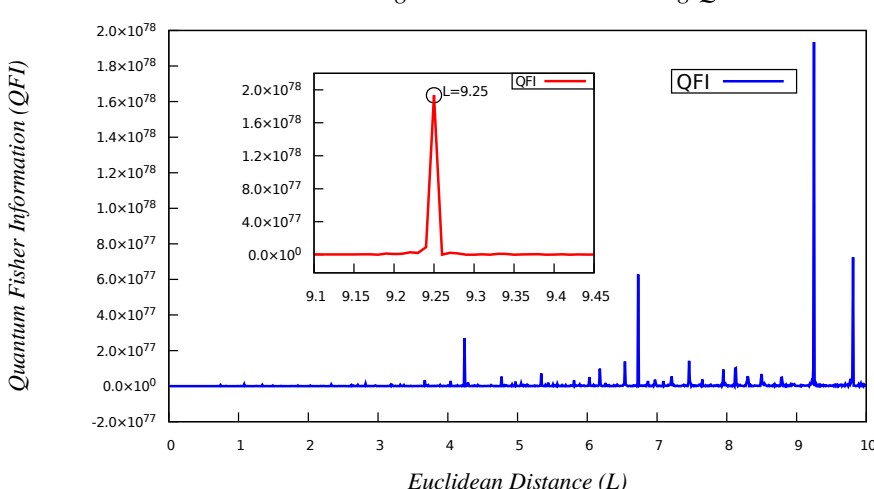

Figure 5: Behavior of the QFI with varying Euclidean Distance (L) between the qubits at : {$\mu$=0.001, k=0.001, $\omega$=1, $\omega_0 = \frac{i}{0.002}$, t=1, $\tau = 100$}. Relevant points are marked in the plot.

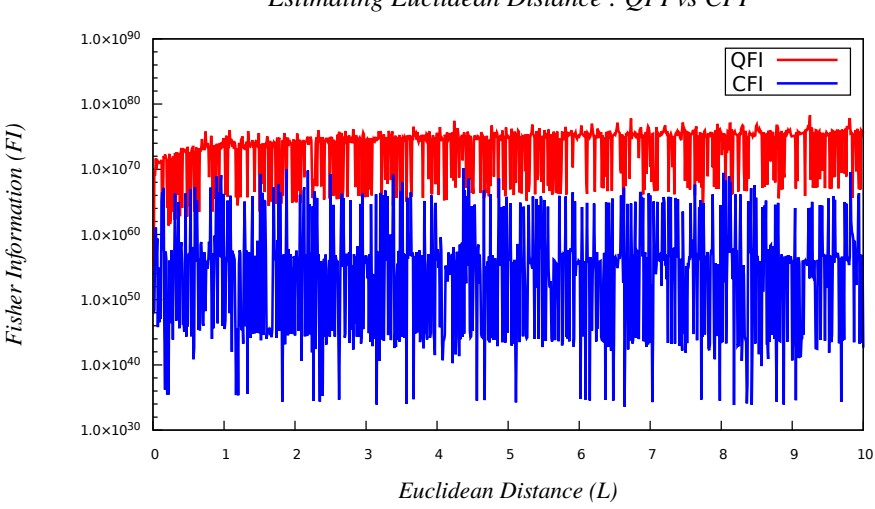

Figure 6: Comparative study of CFI and QFI with varying Euclidean Distance (L) between the two qubits at : {$\mu$=0.001, k=0.001, $\omega$=1, $\omega_0 = \frac{i}{0.002}$, t=1, $\tau = 100$}

distance of about 9.25 between the two qubits as the most appropriate one for studying entanglement related phenomenon between the two qubits. Thus Quantum Fisher Information estimates the euclidean distance between the qubits to be around **9.25** up to certain order of accuracy for these particular choice of other parameter values.

In Fig.(6), we have done a comparison between the QFI and CFI to provide a justification of the fact that QFI is a better way of estimating a parameter characterizing the system than CFI. From the plot it can be seen that QFI provides a better estimation up to 8 orders than CFI in our case for estimating the euclidean distance between the two qubits.

**Sci**|**Post**    SciPost Phys. Core 4, 006 (2021)

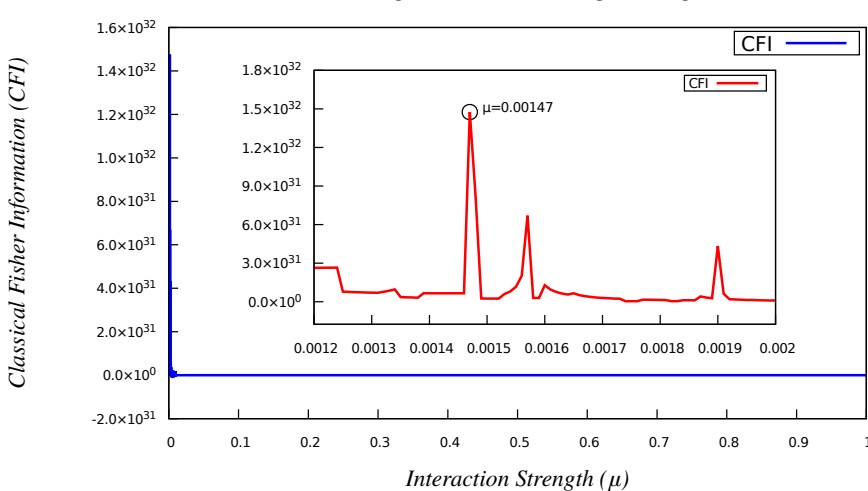

Figure 7: Estimation of coupling strength($\mu$) using the CFI at :{k=0.001, $\omega$=1, $\omega_0 = \frac{i}{0.002}$, L=1, t=1, $\tau = 100$}. Relevant points are marked in the plot.

## 4.3 Estimation of Coupling Strength

The prime objective of here is to estimate the parameter which determines the degree of influence that the environment has on the system. For an open quantum system, the influence of the surroundings on the system can never be neglected. No matter how small the interaction is, it has a significant impact on the time dynamics of the system. The essentiality of this parameter can be understood when one tries to quantify the interaction between the system and the surroundings. This analysis provides estimation of the interaction beyond which it cannot be considered weak and perturbative analysis no longer holds.

In Fig.(7), we have tried to estimate the coupling strength of interaction between our model system and the de-Sitter bath from Classical Fisher Information. From the graph it is very clear that we have obtained peaks for coupling strength parameter less than one, as it should be since we have assumed the weak interaction for Markovian process. The coupling strength less than one means system and bath are weakly coupled and perturbation theory can be applied. From the plot, it is clear that we have a prominent peak at 0.00147 and some other small peaks before 0.002. It is clear that since 0.002 << 1, the perturbation method can be implemented.

In Fig.(8), we plot QFI against interaction strength. It can be seen that we have obtained peaks over a greater range i.e. from 0 to 0.06 of interaction strength. Beyond that we have (almost) no peaks. Beyond establishing the conclusion from the previous graph that our estimated interaction strength has to be small, we observe that QFI is a better estimator of a parameter (here interaction strength) than it's classical counterpart as it provides a wider range for estimation.

The third plot, Fig.(9), is a logarithmic plot in which we plot both CFI and QFI to grasp a better clarity. Both QFI and CFI vary between their maximum value and minimum value. Here also, we can see that QFI is a better estimator than CFI since the range between maximum and minimum for QFI is considerably more than that of CFI. The maximum (and the minimum value) fluctuates highly, but its average value keeps on decreasing gradually with increase in interaction strength. With careful observation it can be noted that the range between maximum and minimum is larger at interaction strength close to zero than the interaction strength close to 1. The wider range near the start gives us a higher chance to obtain a weak cou-

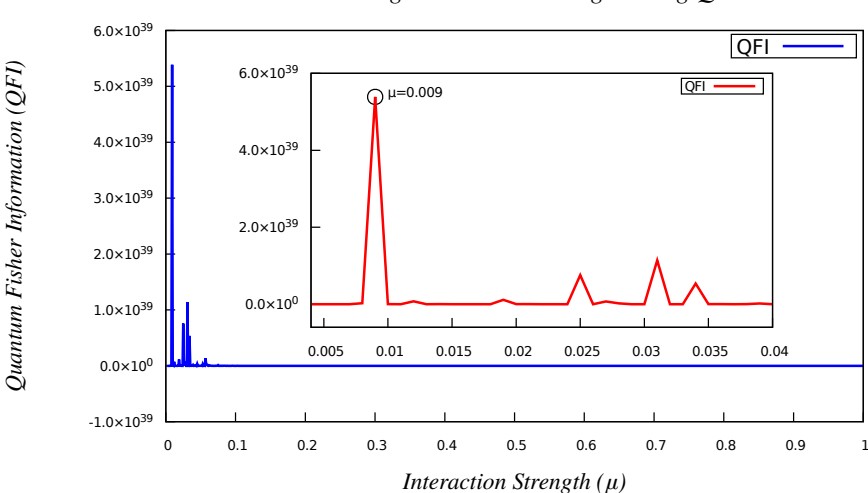

Figure 8: Estimation of coupling strength($\mu$) using the QFI at : {k=0.001, $\omega$=1, $\omega_0 = \frac{i}{0.002}$, L=1, t=1, $\tau = 100$}. Relevant points are marked in the plot.

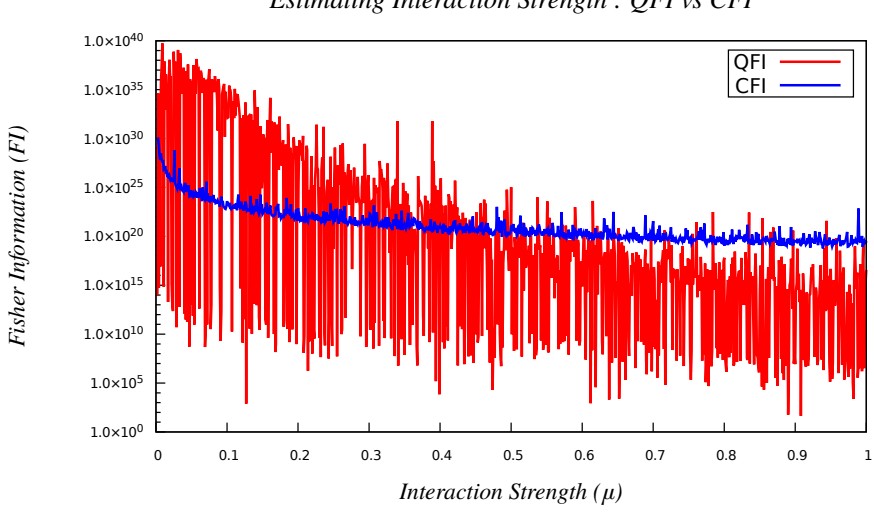

Figure 9: Comparative study between CFI and QFI with changing coupling strength ($\mu$) at : {k=0.001, $\omega$=1, $\omega_0 = \frac{i}{0.002}$, L=1, t=1, $\tau = 100$}

pling strength, again proving our assumption right that our interaction strength should be very small.

# 5 Conclusion

In this paper, we investigated *Quantum Metrological* ideas and techniques in the context of Cosmological background through a simple model. We have used the Fisher Information, both classical and quantum, to estimate certain parameters *viz. Time scale, Euclidean Distance* and *Interaction Strength between the system and the ambient space-time*.

For time scale estimation, we observed that even though CFI and QFI provide more or less same features in early time scale, QFI gives more information about the system in late

| CFI | QFI |
|---|---|
| It is a measure of how quickly a probability distribution changes with parameters. | It represents how quickly a quantum state changes w.r.to some parameters. |
| $\lambda_i$ plays the role of probability. These can be thought as the eigen values of the density matrix. | $\|\lambda_i\rangle$ here represents a state. Eigen vector information is also needed. |
| This can be interpreted as the classical corelation. | This can be interpreted as the quantum entanglement since $\left\|\langle \partial_a \lambda_i \| \lambda_j \rangle\right\|$ illustrates the quantum coherence between the eigenvectors of the density matrix. |
| For our model, we do not see any revival of out of equilibrium features using CFI. | In this case, we obtain a revival of out-of-equilibrium feature at late time scale using QFI. |
| CFI cannot measure the Euclidean distance and the interaction strength as precisely as QFI | QFI estimates the Euclidean distance and interaction strength to a greater order of precision. |

time scales. QFI provides evidence of late time non-equilibrium phenomena being present, for a short period, through a revival mechanism after an initial equilibrium phase between the system and the bath.

By estimating Euclidean distance, we observed that for our model QFI provides a better accuracy than CFI as the peak for QFI occurs at a larger order of magnitude than the CFI. In our case we can say that the entanglement between two spins becomes maximum when the QFI peaks to highest value for the particular value of Euclidean distance.

From our analysis, we found that the interaction strength is estimated to be very less compared to 1 and hence our assumption about weak interaction between system and the bath is justified and the use of perturbative methods permitted.

Following the analysis, we have observed that QFI not only estimates parameters to a better accuracy it does so with a better precision as well. Hence, *QFI is indeed a superior parameter estimator than CFI.*

Here we present few subtle differences between classical and quantum Fisher information.

# Acknowledgments

SC would like to thank Max Planck Institute for Gravitational Physics, Potsdam for providing the Post-Doctoral Fellowship. Satyaki and Abinash would like to thank NISER Bhubaneswar for providing best research atmosphere and fellowship. NG would like to show his gratitude towards IISER Mohali for the continued support. Last but not the least, we would like to acknowledge our debt to the people belonging to the various part of the world for their generous and steady support for research in natural sciences.

# Appendix

## Construction of Effective Hamiltonian and Lindbladian

In the GSKL Master equation we encountered two important terms that involved Effective Hamiltonian and Lindbladian. Here, we express these two terms with some details to help readers. Detailed description of this can be found in [4,5].

The first term in the right hand side of the master equation is governed by constructing the effective Hamiltonian, which describes the unitary part of the time evolution of the two qubit system along with the quadratic interaction between the two qubits after integrating out the contribution from the bath modes. It is given by the following expression.

$$H_{eff} = H_S + H_{LS} = \frac{\omega}{2} \sum_{\alpha=1}^{2} n^{\alpha} \cdot \sigma^{\alpha} - \frac{i}{2} \sum_{\alpha=1}^{2} \sum_{i,j=+,-,3} H_{ij}^{\alpha\beta} (n_i^{\alpha} \cdot \sigma_i^{\alpha})(n_j^{\alpha} \cdot \sigma_j^{\alpha}). \tag{33}$$

Each of the entries of $H_{ij}^{\alpha\beta}$ are computed from the thermal ensemble average of the two-point correlation functions of the massless scalar field $\Phi$ placed at two different coordinate position of the individual qubit and $H_{LS}$ appears from the interaction between atomic system and the environment and is commonly identified as the *Lamb Shift* Hamiltonian, which is frequently used to determine the curvature of the static patch of De Sitter space from the spectroscopic shifts obtained from the four possible entangled quantum states i.e. ground, excited, symmetric and anti-symmetric states of the two qubit system. For two qubit system the *Wightman function* basically appears as a $(2 \times 2)$ matrix, where the diagonal components are same and physically represent the two-point thermal auto correlation function. On the other hand, the off-diagonal components are symmetric under the exchange of the qubit index and give rise to the same expression for the two-point thermal cross correlation functions. Now to explicitly compute these expression one needs to compute the average of the thermal ensemble. But we all know that this operation can be performed by computing the trace operation in presence of a thermal *Boltzmann factor*, $\exp(-\beta H_B)$, where $\beta = 1/T$ (in *Boltzmann constant*, $k_B = 1$ natural unit) in which $T$ represents the equilibrium temperature of the thermal bath. $T$ is characterized by the expression, $T = 1/2\pi k$, where $k$ is a length scale of the theory, which is proportional to the inverse of the square root of the $3+1$ dimensional Cosmological Constant of the static patch of de Sitter space. Here $H_B$ is the bath Hamiltonian. But we all know that performing such thermal trace operation is not allowed in the context of the static patch of de Sitter space as the discrete eigenstate representation of the trace operation do not exist. The prime reason is one cannot treat the present cosmological set up as a fully quantum mechanical experiment which can be performed many times and as a result one cannot write the outcomes in terms of the energy eigenvalues for the present cosmological set up. For this reason we need to extend the thermal trace operation in the finite temperature quantum field theory set up in which by making use the basic principles of the well known *Schwinger-Keldysh Path Integral formalism* one can explicitly compute the expressions for the auto and cross correlation functions in the present context. Additionally, it is important to mention here that, the effective strength of the two qubit quadratic interaction is characterized by the quantity, $H_{ij}^{\alpha\beta}$, which can be obtained by performing the *Hilbert transformation* of the *Fourier transformed Wightman function*. In the context of two qubit system the effective interaction strength of the quadratic interaction is characterized by the following expression:

$$H_{ij}^{(\alpha\beta)} = \begin{cases} \mathcal{M}_1^{\alpha\alpha}(\delta_{ij} - \delta_{3i}\delta_{3j}) - i\mathcal{N}_1^{\alpha\alpha}\epsilon_{ijk}\delta_{3k}, & \alpha = \beta \\ \mathcal{M}_2^{\alpha\beta}(\delta_{ij} - \delta_{3i}\delta_{3j}) - i\mathcal{N}_2^{\alpha\beta}\epsilon_{ijk}\delta_{3k}. & \alpha \neq \beta \end{cases}, \tag{34}$$

with $i, j = +, -, 3$; $H_{ij}^{\alpha\beta} = H_{ij}^{\beta\alpha}$ and we also define:

$$\mathcal{M}_1^{\alpha\alpha} = \frac{\mu^2}{4}\left[\Delta^{(\alpha\alpha)}(\omega_0) + \Delta^{(\alpha\alpha)}(-\omega_0)\right] \approx 0, \tag{35}$$

$$\mathcal{N}_1^{\alpha\alpha} = \frac{\mu^2}{4}\left[\Delta^{(\alpha\alpha)}(\omega_0) - \Delta^{(\alpha\alpha)}(-\omega_0)\right] \approx 0, \tag{36}$$

$$\mathcal{M}_2^{\alpha\beta} = \frac{\mu^2}{4}\left[\Delta^{(\alpha\beta)}(\omega_0) + \Delta^{(\alpha\beta)}(-\omega_0)\right] = \frac{\pi\omega_0}{2}\mathcal{Z}(\omega_0, L/2), \tag{37}$$

$$\mathcal{N}_2^{\alpha\beta} = \frac{\mu^2}{4}\left[\Delta^{(\alpha\beta)}(\omega_0) - \Delta^{(\alpha\beta)}(-\omega_0)\right] \approx 0, \tag{38}$$

where $\mathcal{Z}(\omega_0, L/2)$ is defined later. Here $\Delta^{\alpha\beta}(\pm\omega_0) \forall (\alpha, \beta = 1, 2)$ represent the *Hilbert transformation* of the *Wightman functions* which can be computed as:

$$\Delta^{\alpha\alpha}(\pm\omega_0) = \frac{P}{2\pi^2 i}\int_{-\infty}^{\infty} d\omega \, \frac{1}{\omega \mp \omega_0}\mathcal{G}^{\alpha\alpha}(\omega), \tag{39}$$

$$\Delta^{\alpha\beta}(\pm\omega_0) = \frac{P}{2\pi^2 i}\int_{-\infty}^{\infty} d\omega \, \frac{1}{\omega \mp \omega_0}\mathcal{G}^{\alpha\beta}(\omega), \tag{40}$$

where $\mathcal{G}^{\alpha\alpha}(\omega)$ and $\mathcal{G}^{\alpha\beta}(\omega)$ represent the *Fourier transform* of all the components of *Wightman function*, which are defined as:

$$\mathcal{G}^{\alpha\alpha}(\omega) = \int_{-\infty}^{\infty} d\mathcal{T} \, e^{i\omega\mathcal{T}} \, \mathbf{G}^{\alpha\alpha}(\mathcal{T}) = \frac{\omega}{(1 - e^{-2\pi k\omega})}, \tag{41}$$

$$\mathcal{G}^{\alpha\beta}(\omega) = \int_{-\infty}^{\infty} d\mathcal{T} \, e^{i\omega\mathcal{T}} \, \mathbf{G}^{\alpha\beta}(\mathcal{T}) = \frac{\omega\mathcal{W}(\omega, L/2)}{(1 - e^{-2\pi k\omega})}. \tag{42}$$

Here, the components of the *Wightman functions*, $\mathbf{G}^{\alpha\alpha}(\mathcal{T})$ and $\mathbf{G}^{\alpha\beta}(\mathcal{T})$ represent the two-point auto and cross correlation functions respectively. Here $\mathcal{T} := \tau - \tau'$ represents the time interval. Also in this context, $P$ represents the principal part of the each integrals. For simplicity we also define frequency and euclidean distance dependent two new functions $\mathcal{W}(\omega, L/2)$ and $\mathcal{Z}(\omega, L/2)$ [5] which are given by the following expressions:

$$\mathcal{W}(\omega, L/2) = \frac{\sin(2k\omega\sinh^{-1}(L/2k))}{L\omega\sqrt{1 + (L/2k)^2}}, \tag{43}$$

$$\mathcal{W}^2(\omega, L/2) + \mathcal{Z}^2(\omega, L/2) = \left(L\omega\sqrt{1 + (L/2k)^2}\right)^{-2}. \tag{44}$$

The second term in the master equation is the source of *quantum mechanical dissipation* and describes processes like transition, dissipation and decoherence of the qubit system due to the presence of an external field, which here is the massless probe scalar field $\Phi$ placed at the thermal bath. It is commonly known as the *Lindbladian operator* in the context of OQS and is given by the following expression:

$$\mathcal{L}[\rho_S(t)] = \frac{1}{2}\sum_{\alpha=1}^{2}\sum_{i,j=+,-,3} C_{ij}^{\alpha\beta}[2(n_i^\beta . \sigma_i^\beta)\rho_S(n_i^\alpha . \sigma_i^\alpha) - \{(n_j^\alpha . \sigma_j^\alpha)(n_j^\beta . \sigma_j^\beta), \rho_S\}], \tag{45}$$

---

[5]Here we have introduced two length scales, which are give by, the *Euclidean distance scale*, $L = 2r\sin(\Delta\theta/2)$, where $\Delta\theta$ and $r$ represent the angular separation and the radial distance of the two static qubits and the length scale which is associated with the inverse of the Cosmological Constant, $k = \sqrt{\zeta^2 - r^2} = \sqrt{3/\Lambda - r^2}$ is directly related to the $3 + 1$ dimension Cosmological Constant in the static patch of De Sitter space.

where $\{,\}$ represents the anti-commutation operation between two qubit matrices in the new transformed basis. The explicit mathematical form of this *Lindbladian operator* appearing here is unique in the context of two qubit system. Here, the strength of the quantum dissipation mechanism is characterized by, $C_{ij}^{\alpha\beta}$, which can be expressed in terms of the *Fourier transformation* of the individual components of the *Wightman function*. In the context of two qubit system the effective interaction strength of the quantum dissipation mechanism is characterized by the following expression:

$$C_{ij}^{(\alpha\beta)} = \begin{cases} \widetilde{\mathcal{M}_1^{\alpha\alpha}}(\delta_{ij} - \delta_{3i}\delta_{3j}) - i\widetilde{\mathcal{N}_1^{\alpha\alpha}}\epsilon_{ijk}\delta_{3k}, & \alpha = \beta \\ \widetilde{\mathcal{M}_2^{\alpha\beta}}(\delta_{ij} - \delta_{3i}\delta_{3j}) - i\widetilde{\mathcal{N}_2^{\alpha\beta}}\epsilon_{ijk}\delta_{3k}. & \alpha \neq \beta \end{cases}, \tag{46}$$

with $i, j = +, -, 3$; $C_{ij}^{\alpha\beta} = C_{ij}^{(\beta\alpha)}$ and we also define:

$$\widetilde{\mathcal{M}_1^{\alpha\alpha}} = \frac{\mu^2}{4}\left[\mathcal{G}^{(\alpha\alpha)}(\omega_0) + \mathcal{G}^{(\alpha\alpha)}(-\omega_0)\right] = \frac{\mu^2\omega_0}{8\pi}\coth(\pi k\omega_0), \tag{47}$$

$$\widetilde{\mathcal{N}_1^{\alpha\alpha}} = \frac{\mu^2}{4}\left[\mathcal{G}^{(\alpha\alpha)}(\omega_0) - \mathcal{G}^{(\alpha\alpha)}(-\omega_0)\right] = \frac{\mu^2\omega_0}{8\pi}, \tag{48}$$

$$\widetilde{\mathcal{M}_2^{\alpha\beta}} = \frac{\mu^2}{4}\left[\mathcal{G}^{(\alpha\beta)}(\omega_0) + \mathcal{G}^{(\alpha\beta)}(-\omega_0)\right] = \frac{\mu^2\omega_0}{8\pi}\coth(\pi k\omega_0)\mathcal{W}(\omega_0, L/2), \tag{49}$$

$$\widetilde{\mathcal{N}_2^{\alpha\beta}} = \frac{\mu^2}{4}\left[\mathcal{G}^{(\alpha\beta)}(\omega_0) - \mathcal{G}^{(\alpha\beta)}(-\omega_0)\right] = \frac{\mu^2\omega_0}{8\pi}\mathcal{W}(\omega_0, L/2), \tag{50}$$

where $\mathcal{G}^{\alpha\alpha}(\omega)$ and $\mathcal{G}^{\alpha\beta}(\omega)$ represent the *Fourier transform* of the all components of *Wightman function* defined earlier.

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
