# Peer review of "QMetrology from QCosmology: Study with Entangled Two Qubit Open Quantum System in De Sitter Space"

_SciPost Physics Core, doi:SciPost Phys. Core 4, 006 (2021)_

## Round 1 · Referee Report · Anonymous (Referee 1) · 2021-1-3

Weaknesses

  1. As far as I understood, the authors are studying a quantum system. Hence it is a little bit confusing to me when they talk about using classical Fisher information and compare it with quantum Fisher information. They should clarify this point.

  2. The definition of quantum Fisher information matrix given in equation (39) of the paper. Then for a using the spectral decomposition of the density matrix of the system it the takes the form mentioned in equation (47) and various terms in (47) are mentioned in (48), (49) and (50). The part mentioned in (48) is termed as "Classical Fisher Information." In equation (38), another definition of classical Fisher Information is given. How do these two things correlate?

  3. Furthermore, in the subsequent section, the authors claimed that they had computed both the classical and quantum Fisher Information. I believe they have used (48) to compute the classical Fisher Information. Then where the discussion of section A of page-6 of the paper enters into the picture? If not, I will suggest removing that from the paper.

  4. If what I mentioned in the previous point is correct then, there is no reason to only compute the first term (the classical part) of the equation (47) and then compare with the full expression. More clarifications are needed in this regard.

  5. Last but not least, the significance of the result is not clear at all. There are already various works on Quantum Fisher Information where it is used to study the dynamics of a quantum system ( some of them are already cited in the paper itself) and its usefulness is also discussed. So what new things we learn from this exercise about the underlying system? Impact of the result mentioned in the paper is not very clear to me. Authors should expand upon on it.

Report

Authors used Quantum Fisher Information for a two-qubit system coupled with a thermal bath. The bath is modelled by a conformally coupled scalar field on a background of de Sitter spacetime. They observe a revival while studying the system dynamics. The calculation seems to be reasonable, but the significance of the result, especially in the context of the underlying result is not very clear. Do we learn anything novel about the underlying system apart from observing the revival? Why this particular model is chosen ?. To me the significance and impact of the result is not sufficient enough to merit a publication in Sci-Post physics, I recommend this for publication in Sci-Post Physics core instead after clarifying the doubts.
  • validity: ok
  • significance: low
  • originality: low
  • clarity: ok
  • formatting: reasonable
  • grammar: reasonable

Author:  Sayantan Choudhury  on 2021-01-19  [id 1163]

(in reply to Report 1 on 2021-01-03)

Response to the referee is attached as .pdf file and submitted.

Attachment:

Fish_info_ref_report2_JRkcN9U.pdf

---

## Round 1 · Referee Report · Anonymous (Referee 2) · 2021-1-7

Report

In the paper the authors investigate the quantum metrological properties in dS space. They evaluate the attainable metrological gain for a 2 qubits system coupled to a thermal bath by using the Cramer-Rao bound of the Fisher information, both classical and quantum version. The authors discuss the performance of both Fisher information on few quantities such as the following terms I quoted directly from the paper: "Time scale, Euclidean Distance and Interaction Strength between the system and the ambient space-time."

The central idea of this paper is that one can still attain some useful information for metrological purpose even when coupled to bath is interesting. However, this idea has already been well studied in numbers of system such as spin squeezing using optical cavity. Except for the dS space that the authors discussed, I don't see any innovation in this manuscript. Also, the whole story of QFI is that people know there exists a way to gain information, but what actually matters is HOW to extract them. The authors did not discuss anything about it.

Speaking of the dS space, I don't see any reasons to study in this particular spacetime. Also, the paper is trying to understand the problem using quantum mechanics in dS space, which was not commonly used and the validity should be justified.

Some of the discussed quantities are barely explained.

There are also certain amount of formatting and writing problems. - I was very lost in page 2-5; for example in page 4 it took me a while to realize that the bottom left part is the unfinished footnote in page 3. I think the authors should not write so long footnotes: if the contents are important please move them to the main text; otherwise please remove them. - Also, I recommend the authors to remove the review of Fisher information and leave the necessary references should be enough. - The authors should add paragraphs about quantum metrology in the introduction part. - And the introduction in its current form contains too many unnecessary terms and abbreviations which are irrelevant to the whole paper, making it unreadable. In fact, there are way too many introduced concepts all over the paper, which are barely used later in the paper.

Requested changes

See report.

  • validity: ok
  • significance: low
  • originality: low
  • clarity: low
  • formatting: acceptable
  • grammar: below threshold

Author:  Sayantan Choudhury  on 2021-01-19  [id 1162]

(in reply to Report 2 on 2021-01-07)

Response to the referee is attached as .pdf file and submitted.

Attachment:

Fish_info_ref_report2.pdf

---

## Round 3 · Referee Report · Anonymous · 2021-2-3

Weaknesses
1. The analysis appears to be erroneous.
2. It is not clear why the problem studied here is significant.
3. Very similar studies have appeared in the past.
4. Relevant past works and similar studies have not been cited.
5. The paper does not always clearly distinguish what has been done by others and what is the original contribution of the authors.
6. Presentation lacks clarity.
Report
This paper studies Fisher Information for a particular system: a pair of qubits coupled to a scalar field which acts as a thermal bath. The body of the paper starts with a section on the 'two qubit open system' which (though it was not clearly stated) is a review of previous work. At the end of this section, schematic formulae for the density matrices of the two qubits are given and it is stated that the detailed expressions have been calculated in a different paper (co-authored by some of the authors) in the approximation $2 \pi k \omega >> 1$ and subject to the constraint $\coth (\pi k \omega_0)=0$ where $k, \omega,\omega_0$ are all parameters that enter the density matrices of the qubits. It is stated that the results of this other paper ( not explicitly given here) will be used in obtaining Fisher information.
The original contribution of the paper begins from the section titled 'Estimation of Parameters' and essentially consists of using the density matrices mentioned above to calculate classical and quantum Fisher Information for this system. Variation of CFI and QFI with the different parameters that enter the density matrices is studied.
Now let us elaborate on the issues with the paper:
1. The analysis appears to be erroneous. As per the statement made in the paper, the results for density matrices were obtained in the para regime $k\omega \gg 1 $ subject to the constraint $\coth \pi k \omega_0 =0$. Now there is no real solution to the constraint, the only solution is $k \omega_0 = (n+1) i/2 $ for integer $n$. $\omega_0$ being real, this means $k$ must be purely imaginary. Then it is unclear how $k \omega \gg 1 $ can be fulfilled, since $\omega$ is real again. In any case, in the section on Fisher Information only real values of all the parameters have been considered and further only small values $k\omega$ have been considered. So it appears that neither of the two conditions, under which the paper claims to have derived the expressions for density matrices, were satisfied when the same density matrices were used for the calculation of Fisher Information.
2. It is not clear why the problem studied here is significant: Even if the above issues did not exist, the study here would still need to establish significance. It has not been established why the qubit system is of physical interest, or why metrological considerations are appropriate for this system, or if the calculation of QFI tells us anything non-trivial. There have been interesting applications of Fisher Information in a de Sitter background (for instance, to ask if there are fundamental limitations to measuring cosmological observables of interest), but in this case the study of QFI appears to be arbitrary.
3. Similar studies have appeared in past: Quantum Fisher information for an Unruh-DeWitt detector coupled to a scalar field in a dS background was obtained in arXiv 1806.08922, QFI for a qubit coupled to a scalar field in a dS background was obtained in 'Protecting quantum Fisher information in curved space-time' by Zhiming Huang (EPJP, 2018). Even if significance could be established for this direction of research, the paper would have to be establish how this study is telling us something significantly different from these previous ones.
4. Relevant past works and similar studies have not been cited: the two papers mentioned above which also studied QFI in a very similar context have not been cited. The model of the two qubit system in de Sitter used here first appeared in arXiv 1310.7650 and the formulae given in this paper appear to closely follow the results presented there. While this paper has been cited in previous papers by some of the same authors, it was not cited here. Some other relevant references which studied Fisher information or the dynamics of similar systems in de Sitter space but have not cited: 1812.02345, 1707.09702, 1407.4930, 1605.07350, 1706.0917, 1707.08414.
5. The paper does not always clearly distinguish what has been done previously by others and what is the original contribution of the authors: The authors did not mention where the qubit model used here originated. The fact that it had already been studied in de Sitter space (first in 1310.7650) should have been mentioned. These omissions, together with the authors' referring to the qubit model as 'our model', can be misleading for the reader. More generally, there is no discussion in the introduction about any previous work on open systems in dS (even though some of this research is cited), which can again give an incorrect impression about the originality of the paper.
6. Presentation lacks clarity: Previous results that have been used as the main input in the study of this paper were not presented explicitly nor was a detailed reference given (see point 1 above). Quantities were not defined where they were introduced (for instance, $L$ and $\omega_0$ first appear in eqns 10 and 11 and are not defined till two pages later). There were a number of things that were not clearly explained (such as the distinction between $\omega$ and $\omega_0$, both of which are defined as 'Fourier modes of the Wightman functions' in the paper). Lack of paragraph breaks hampered readability (the whole first column of the second page is a single block of text).
Author: Sayantan Choudhury on 2021-03-16 [id 1312]
(in reply to Report 1 on 2021-02-03)We have attached the response to the referee report as .pdf file. Please look into this.
Anonymous on 2021-03-01 [id 1273]
I'm the referee of "Anonymous Report 2 on 2021-2-23 Contributed Report", and the [1] reference in my report is Nature Physics 7, 406–411(2011), by B. M. Escher, R. L. de Matos Filho & L. Davidovich.
Anonymous on 2021-02-03 [id 1199]
This is an erratum by the referee of report 1. We mistakenly wrote in point (1) that $\coth k\omega_0 = 0$ implies $k \omega_0 =\infty$ when actually, there is no real value of the arguments for which that equation is satisfied. This is an even stronger objection to the results here, which are claimed to have been derived assuming $\coth k\omega_0 = 0$ for real parameters $k,\omega_0$. It shows that there is no parameter range for which the approximation used is valid.
Anonymous on 2021-02-02 [id 1196]
I noticed the following statement in the text which appears to be inaccurate:
"Scientists have used qubits oriented along z-axis as the primary objects when they talk about entanglement. Every student once in his student life has asked himself ‘why do we study the qubits directed along z-axis mainly?’ since literature lacks the study of entanglement where atoms are oriented in direction other than z-axis."
It seems to be saying that the z-axis is given a special treatment, or there is some loss of generality involved in taking a pair of spins along z-axis. Which, because of isotropy, is not true. Perhaps the authors meant something else, in which case I would suggest reframing the sentence.
Anonymous on 2021-02-02 [id 1197]
(in reply to Anonymous Comment on 2021-02-02 [id 1196])The authors appreciates the person for this useful comment. We will be careful to reframe the sentence in the future version of the paper.

---

## Round 3 · Referee Report · Anonymous · 2021-2-23

Report
The authors have improved a lot on the formatting which makes me satisfied now. However I still need more explanations about the following major parts before this paper going any further:
1. Why is the studied object important? As I also mentioned in the last round of review, dS space does not bring anything noteworthy and therefore the authors must underscore why should we care about this particular problem: does it bring new understanding of quantum metrology (like breaking the no-go theorem of quantum metrology with dephasing, see [1] below), or this particular space is commonly encountered?
2. For such a simplified system, the author should discuss about what is the best way to fully use the Fisher information in order to estimate the phase. For large scale system this remains an open question, but for 2-qubit system this should be derived and included in the text.
Also, as pointed out by other referees, I recommend the authors to clearly distinguish introduction & review with the original works derived in this paper. Current form of this manuscript is still ambiguous to me.
Author: Sayantan Choudhury on 2021-03-16 [id 1311]
(in reply to Report 2 on 2021-02-23)Please look into the attached .pdf file where the response to the referee's comments are attached.
Attachment:

---

## Round 3 · Referee Report · Anonymous · 2021-2-28

Report
Dear Editor,
Authors have answered some of the questions raised by me. It can be accepted for publication.
Author: Sayantan Choudhury on 2021-03-16 [id 1310]
(in reply to Report 3 on 2021-02-28)Please look into the attached file where response is written.
Attachment:

---

## Round 3 · Author Response

Here we are submitting the updated version of the draft by following both of the referees valuable suggestions. We are thankful to the referees for giving us extremely significant inputs/comments, which helped us to improve the presentation of the revised version of the manuscript. We are thankful to the Editor for communicating with us many times and helped us to resolve various technical issues regarding submission. We believe that the revised version have considerably addressed the crucial issues as asked by the referees. In this regard, we request the Editor to consider this version for the publication in the prestigious journal SciPost.
Best regards,
Dr. Sayantan Choudhury and the team.

---

## Round 3 · List of Changes

The list of changes in the revised version are already pointed in the response to the referee report 1 and 2.

---

## Round 4 · Author Response

Dear Editor,

Thank you very much for the previous correspondence and also we want to thank all the referees for giving us their valuable comments. We are thankful to one of the referees for accepting our previous version. Though there us objection by one new referee and another referee have asked few explanations. By following the details of both the reports we have sufficiently modified the revised version of the draft which we have uploaded in arXiv as well. In this regard, I would like to request the Editor-in-Charge to reconsider our paper for publication in SciPost.

Best regards,
Dr. Sayantan Choudhury (Corresponding Author)

---

## Round 4 · List of Changes

We have added the following changes: 1. We have provided all the explationations to the objection of one of the referees. 2. We have added discussion regarding to motivation and novelty of the work performed which was asked by another referee. 3. We have added new citations as suggested by one of the referees.

---

## Editorial Decision

published